# Enhanced Outer Membrane Vesicle Production in *Escherichia coli*: From Metabolic Network Model to Designed Strain Lipidomic Profile

**DOI:** 10.3390/ijms26146714

**Published:** 2025-07-13

**Authors:** Héctor Alejandro Ruiz-Moreno, Juan D. Valderrama-Rincon, Mónica P. Cala, Miguel Fernández-Niño, Mateo Valderruten Cajiao, María Francisca Villegas-Torres, Andrés Fernando González Barrios

**Affiliations:** 1Grupo de Diseño de Productos y Procesos (GDPP), Chemical and Food Engineering Department, Universidad de los Andes, Bogotá 111711, Colombia; ha.ruiz75@uniandes.edu.co (H.A.R.-M.); mafernandezn@gmail.com (M.F.-N.); 2GRESIA Group, Environmental Engineering Faculty, Universidad Antonio Nariño, Bogotá 111511, Colombia; juan.valderramar@uan.edu.co; 3Metabolomics Core Facility—MetCore, Vicepresidency for Research and Creation, Universidad de los Andes, Bogotá 111711, Colombia; mp.cala10@uniandes.edu.co; 4Department of Biotechnology, Institute of Agrochemistry and Food Technology (IATA-CSIC), 46980 Paterna, Spain; 5Centro de Investigaciones Microbiológicas (CIMIC), Department of Biological Sciences, Universidad de los Andes, Bogotá 111711, Colombia; m.valderruten@uniandes.edu.co (M.V.C.); m.fvillegastorres@uniandes.edu.co (M.F.V.-T.)

**Keywords:** outer membrane vesicles, lipidomics, CRISPR, metabolic network models, *E. coli* JC8031

## Abstract

Bacterial structures formed from the outer membrane and the periplasm components carry biomolecules to expel cellular material and interact with other cells. These outer membrane vesicles (OMVs) can encapsulate bioactive content, which confers OMVs with high potential as alternative drug delivery vehicles or as a platform for novel vaccine development. Single-gene mutants derived from *Escherichia coli* JC8031 were engineered to further enhance OMV production based on metabolic network modelling and in silico gene knockout design (ΔpoxB, ΔsgbE, ΔgmhA, and ΔallD). Mutants were experimentally obtained by genome editing using CRISPR-Cas9 and tested for OMVs recovery observing an enhanced OMV production in all of them. Lipidomic analysis through LC-ESI-QTOF-MS was performed for OMVs obtained from each engineered strain and compared to the wild-type *E. coli* JC8031 strain. The lipid profile of OMVs from the wild-type *E. coli* JC8031 did not change significantly confirmed by multivariate statistical analysis when compared to the mutant strains. The obtained results suggest that the vesicle production can be further improved while the obtained vesicles are not altered in their composition, allowing further study for stability and integrity for use in therapeutic settings.

## 1. Introduction

Using secretincan bioactive compounds, Gram-negative bacteria are able to communicate, relieve stress, and attack foreign entities [1,2]. Proteins, nucleic acids, and compounds found in the periplasm are packaged into spheres made of their own outer membrane (OM) material. Their cargo, protected during travel by the lipid bilayer, can reach host organisms more reliably than compounds secreted directly to the environment. Proteins displayed on the OM can become a part of outer membrane vesicles (OMVs) and provoke adherence with other organisms [3]. Manipulation of membrane proteins and OMV cargo can be leveraged to target selected organisms and inject a desired bioactive material [4,5].

While the effectiveness of widely used antibiotics is declining, naturally occurring phenomena continue to inspire antibiotic therapies, alternative antibiotic sources, and drug delivery [1]. Outer membrane vesicles (OMVs) produced by *Pseudomonas aeruginosa* exhibit antibacterial properties, attacking other bacteria through encapsulated periplasmic autolysins [6]. *Lysobacter* spp. have also been shown to release OMVs with similar bioactive capacities, attributed to compounds within their OMV cargo [7]. The potential for high OMV production is not limited to these species, as demonstrated by other Gram-negative bacteria like *Escherichia coli* [8]. This presents an opportunity to use *E. coli* as a platform for the production of OMVs for drug delivery, the development of novel vaccines [9], or to emulate an OMV-based antibacterial system.

Several factors affect the production of OMVs. Mechanisms affecting their biogenesis are related to deformations in the OM including: (1) disrupted links between the inner membrane, the peptidoglycan layer, and the outer membrane; (2) accumulation of periplasmic proteins; and (3) curvatures induced by OM proteins [2,3,9,10]. Kulp et al. suggest that outer membrane vesicle (OMV) production is influenced by both structural characteristics of the periplasm and the metabolic pathways responsible for synthesizing key envelope components, such as membrane proteins, lipopolysaccharides (LPS), and enterobacterial common antigen (ECA) [11]. Additionally, stress response pathways are reported to affect OMV formation. However, the precise metabolic routes governing OMV biogenesis are not yet fully understood. To address this gap, metabolic network models provide a valuable framework. These models connect genomic data to enzymatic reactions, describing the metabolic states and biological potential of an organism, and can thus help identify conditions or compounds that enhance in vivo OMV production. Further exploration of the effect of gene editing in the metabolic space can increase our understanding of OMV formation and improve its yield.

Being able to produce OMVs in large quantities is an important target. Additionally, the activity and stability of OMVs are important when considering their therapeutic use [12]. Proteins in the cargo and displayed on the membrane affect their bioactivity [3]. Toxic LPS in the outer membrane can also affect their medical use [13]. The OMVs proteome analyses of various Gram-negative bacteria, including *E. coli*, have been reported and lipidomic profiles for Klebsiella pneumoniae OMVs [14,15,16,17] have given an insight into their composition. However, there are currently no lipidomic profiles reported for *E. coli* OMVs.

This study involved adapting a genome-scale metabolic network model of *Escherichia coli* K-12. Subsequently, constraint-based optimization methods were utilized to identify strategies that enhance outer membrane vesicle (OMV) production in *E. coli* JC8031. Description of the vesicle production was achieved through the model constraints and the objective functions driving the optimization problem. We found four *E. coli* JC8031 mutant strains with enhanced vesicle production through in silico gene knockouts. To our knowledge, this is the first use of metabolic network models to study OMVs production and the first report on the lipid composition of isolated *E. coli* OMVs.

In silico modeling was experimentally validated by using a CRISPR-Cas9 platform for in vitro gene deletion. Experimental recovery of OMVs and subsequent lipidomic analysis produced novel results on the lipid profile of *E. coli* OMVs and the effect of mutations on their lipid composition. The integration of computational and experimental approaches through gene editing to enhance OMVs biogenesis are a step forward for high yield production of OMVs for medical applications.

## 2. Results

### 2.1. Refined Metabolic Network Model for E. coli JC8031

We decided to use the *E. coli* K12 metabolic network model iML1515 [18] as a base and adapt it to obtain a computational model that adequately describes vesicle production in *E. coli* JC8031 strain. We initially accounted for the JC8031 strain-specific deletion ΔtolRA [19]. We implemented GrowMatch [20] to verify that in silico results were consistent with cell growth and vesicle production reported in experimental data of 150 samples from the Keio collection [11]. Implementation of GrowMatch resolved modeled growth inconsistencies in 16 instances and helped calibrate the in silico predictions to cell growth data by including additional reactions in the carbon and lipid metabolism or additional constraints. Further additions were made to model protein secretion of proteins to the periplasm through transport or exchange reactions. Model data are available in Appendix A. The employed model consisted of 2160 reactions, 724 transport and exchange reactions, and 2057 metabolites divided into two separate compartments (cytosol and periplasm) along with the extracellular medium.

The objective function employed in the constraint-based methods resulted from the metabolites involved in the formation of the OM in the bacterium, which is further incorporated in the vesicle [1,3,21]. This included phosphatidylethanolamines (PEs) and phosphatidylglycerols (PGs) as well as the core oligosaccharide lipid A (COLIPA) as a precursor to LPS. The objective function was defined by reactions necessary for the synthesis and translocation of membrane material to the periplasm.

### 2.2. E. coli JC8031 Mutant Strains Were Designed In Silico and Constructed In Vitro Using CRISPR-Cas9

We performed a strain design analysis with the objective of identifying mutations that result in increased vesicle production as described by the objective function. We used OptKnock (https://github.com/opencobra/COBRA.tutorials/blob/master/design/optKnock/tutorial_optKnock.m, accessed on 1 May 2025) [22] to find gene knockouts that could be conducive to a greater vesicle production. The in silico strain design was performed by implementing OptKnock using the refined model of *E. coli* JC8031 as input, the developed objective function for vesicle production, and the exchange reactions that accounted for the LB culture media. OptKnock implementation resulted in sets of up to four gene knock-outs that increased the value of the vesicle production objective function while maintaining cell growth levels. A set of four gene knock-outs was obtained after filtering the results (Table 1). A combination for the deletion of these four genes was found to improve vesicle production in silico. Additionally, single-gene knock-outs were also found to produce an enhancing effect in silico. The gene deletions proposed by the OptKnock algorithm were prioritized for their predicted enhancement of OMV production while preserving biomass growth. Although these predictions were computationally driven, we further investigated possible biochemical or mechanistic rationales that could support their impact on vesiculation. *poxB* encodes a peripheral membrane-associated pyruvate dehydrogenase involved in the conversion of pyruvate to acetate. This enzyme is known to associate with the inner leaflet of the cytoplasmic membrane and is influenced by lipid interactions. Its deletion could alter the pool of acetyl-CoA-derived precursors and reduce flux toward acetate overflow pathways, potentially redirecting carbon toward membrane biosynthesis. Moreover, perturbation of membrane-bound enzymes can disrupt envelope homeostasis and indirectly trigger vesicle formation as part of a stress response mechanism. *gmhA* encodes phosphoheptose isomerase, a key enzyme in the biosynthesis of ADP-L-glycero-D-manno-heptose, a critical precursor for the inner core of LPS. Deletion of *gmhA* leads to LPS truncation, particularly the absence of heptose residues, which weakens outer membrane integrity. This structural destabilization is a known inducer of OMV formation, as the loss of LPS heptoses perturbs interactions between LPS and outer membrane proteins or peptidoglycan, facilitating spontaneous vesiculation. Previous studies have reported hypervesiculation phenotypes in *gmhA* and other LPS biosynthesis mutants, reinforcing this mechanistic link. *sgbE* encodes L-ribulose-5-phosphate-4-epimerase, part of the ascorbate utilization pathway, which is involved in pentose metabolism. While not directly related to envelope biogenesis, deletion of *sgbE* could affect the availability of nucleotide sugar precursors or metabolic flux through the pentose phosphate pathway, indirectly influencing lipid biosynthesis or redox homeostasis. It is possible that metabolic shifts arising from this mutation create imbalances in membrane synthesis or stress responses that promote OMV release. *allD* encodes ureidoglycolate dehydrogenase, an enzyme involved in the catabolism of purines and nitrogen recycling via allantoin degradation. Although this function is distant from membrane dynamics, it could impact cellular nitrogen sensing and carbon–nitrogen balance, both of which are known to influence outer membrane composition and stress signaling pathways. Alterations in nitrogen metabolism may affect the expression of envelope stress regulators or the synthesis of nitrogen-containing lipids and phospholipids, thereby modulating OMV biogenesis. Taken together, while not all targets have a direct structural role in OMV formation, their deletion may disrupt global metabolic and envelope homeostasis in ways that trigger or amplify vesicle production. These hypotheses, derived from biochemical understanding, complement the computational predictions and provide a mechanistic foundation for selecting the knockouts for experimental validation.

Given the enhanced vesicle production found in silico, we constructed four *E. coli* JC8031 mutants with each single-gene deletion to experimentally assess the increased vesicle production. Four single deletion mutants (ΔpoxB, ΔsgbE, ΔgmhA, and ΔallD) were obtained by genome editing using CRISPR-Cas9. PCR results confirmed the complete deletion of the selected gene in each of the four mutants.

### 2.3. Constructed Mutants Presented an Enhanced Vesicle Production

We recovered OMVs produced by the four constructed mutants and JC8031 strain in cell cultures to determine the impact of the gene deletions on the total vesicle yield. Vesicle recovery was quantified through UV-absorbance in a wavelength spectrum ranging from 200 to 320 nm [23]. The area under the absorbance spectrum curve (A.U.C) was used as a measure of vesicle quantity and concentration in each sample. Vesicle production varied within sample groups and showed a distinction in the vesicle production between the wild-type *E. coli* JC8031 and the four designed mutant strains (Figure 1A). All four single deletion mutants surpassed the *E. coli* JC8031 vesicle production. Three single deletion mutants displayed a seven-fold increase in recovered vesicle quantity, while deletion of gene ΔsgbE produced 10 times as much as JC8031 strain. The protein concentration in the samples was evaluated using the Warburg–Christian equation [24]. Protein concentration was significantly higher in three of the single deletion mutants, ΔpoxB, ΔsgbE, and ΔallD (Figure 1B). In order to determine if there was an indirect effect of growth rate, we measured it for all mutants, finding no significant effect of the mutation. Also, we carried out SDS-PAGE for all mutants finding two additional bands for ΔgmhA, possibly explaining the differences explained in the following results (Appendix A).

### 2.4. Lipidomic Analysis Shows Knock-Out Strains Keep a Stable Composition

Given that significant changes were found in vesicle production when knocking out selected genes, a lipidomic analysis was performed to assess the possible impacts of these knock-outs on the molecular composition of the OMVs (see Appendix A). Figure 2 shows a summary of all the detected changes in lipid composition as measured by LC-ESI-QTOF-MS. Importantly, depending on the specific analysis and metabolite annotation strategy, the lipids that can be tracked will change [25], so in this case, special attention should be paid to changes in the PEs content, taking into account that these are one of the prevalent lipid compounds expected to be present in OMVs obtained from *E. coli* [26].

OMV lipidomics was characterized by LC-ESI-QTOF-MS for WT strain JC8031 and four single deletion mutants. The average lipid composition across all OMV samples consisted mainly of glycerolipids (~52%), glycerophospholipids (~12%), polyketides (~6%), fatty acyls (~5%), and prenol lipids (~4%). Other lipid components in minor quantities consisted of sphingolipids (<1%), saccharolipids (<1%), and sterol lipids (<1%). Additionally, other OMV compounds remained unidentified (~20%) using available lipid databases and spectral matching tools. This may be due to limitations in current bacterial lipid libraries, the presence of novel or modified lipids specific to OMV biogenesis, or ionization-related challenges during MS detection.

The lipid profile of OMVs from single deletion mutants was compared to OMVs from the WT *E. coli* JC8031. Figure 2 shows the qualitative change in terms of lipid group content. OMVs from strain ΔgmhA showed a decrease in glycerolipids compared to WT JC8031. OMVs from strain ΔpoxB exhibited an increase in fatty acyl, polyketides, and glycerophospholipid participation in the lipid profile. The lipid profile of OMVs from the other single deletion mutant strains Δ*sgbE* and Δ*allD* did not show evident differences with OMVs from the JC8031 strain.

It is worth mentioning that the computational algorithms used for the strain design focused on maximizing a linear combination of the fluxes of the metabolic reactions involving the production of PGs, PEs, and lipid A. A preliminary analysis of the qualitative changes showed that PEs had increased production in three of the single deletion mutants. PGs did not have a fold-change greater than 3 in any of the single deletion mutant strains.

On the other hand, to obtain quantitative data, an unsupervised principal component analysis (PCA) was implemented to analyze the stability of the lipidomic analytical system and the unsupervised distribution of analyzed samples. The quality control samples appear grouped, evidencing the stability of the analytical system during the analysis of all samples (Figure 3A). Then, a discriminant PLS-DA analysis was built to maximize and inspect the differences between the study groups, obtaining no model (negative values of Q2 and CV-ANOVA with *p*-value = 1) between the OMVs sample groups (Figure 3B). No lipids were found to significantly differentiate groups in either multivariate or univariate analysis, so it can be suggested that there are no significant changes in OMVs lipids composition obtained from the knock-out strains when compared to OMVs from *E. coli* JC8031.

## 3. Discussion

Deletion analysis of the refined model revealed that gene knock-outs enhanced vesicle production, as represented by the objective function. Notably, the gene deletions identified by the OptKnock algorithm were not directly associated with phospholipid metabolism. However, two genes had a direct relation to the outer membrane (poxB, gmhA). poxB encodes for a membrane protein that acts as a pyruvate dehydrogenase. This protein is related to membrane binding and presents interactions with lipids [27]. gmhA encodes for a phosphoheptose isomerase involved in the lipopolysaccharide biosynthesis and ΔgmhA have shown lipopolysaccharides lacking heptose [28]. The involvement of these genes with the formation of OM components supports previously proposed mechanisms of vesicle biogenesis through perturbation of membrane proteins, induced curvatures, and membrane integrity [3]. However, the results show a lack of significance in a ΔgmhA alteration over the protein concentration. The other two deleted genes (allD, sgbE) showed an indirect relation to vesicle production. sgbE encodes for an enzyme from an L-ascorbate utilization operon and is involved in the metabolism of nucleotide and aromatic amino acid precursors [29]. The allD gene encodes for an enzyme involved in the assimilation of allantoin and allantoin use as a nitrogen source [30]. These single deletion targets are also indirectly related to vesicle production through the alteration of biomass growth. The use of the Warburg–Christian equation through spectrophotometric methods shows limitation to quantify vesicle production. This method can be complemented with LPS measurements through the Purpald test [24].

In silico deletion experiments indicated that while the parental strain exhibited a high biomass growth rate, multiple gene deletions did not reduce this rate to a point that would compromise the engineered strain’s utility. Potential discrepancies between the in silico predictions and in vitro outcomes could be observed by experimentally implementing the suggested gene deletions. This implies a challenge in generating multiple subsequent deletions and achieving the expected biomass growth. Here, we found that deletions associated with a reduction in cellular growth also showed a reduced yield for vesicle recovery.

On the other hand, from the industrial point of view, biomass growth was not significantly affected for all mutants while OMVs generation was enhanced up to 10 times in the single deletion mutant strain ΔsgbE, therefore, more attractive when scaling up the process from an economical perspective.

The composition of the produced OMVs is also important when considering their use in therapeutic applications. Previous studies have raised concern on the effect of the recovery method on the size distribution, morphology, and composition of OMVs. However, a comparison of isolation methods has shown that different methods produce a similar effect on vesicle morphology [5]. The lipidomic analysis showed no statistically significant change in the composition that separated the single deletion mutants from the JC8031 strain. This gives an insight into the response of general OMVs composition with respect to genome alteration. The lack of change in the lipid profile could be attractive for biomedical applications as it facilitates the study and analysis of structure stability of OMVs as envelopes for the desired cargo and the effect of external factors on their structural integrity.

To the best of our knowledge, the studies reporting OMVs composition are relatively limited, and they are mostly focused on proteomics analysis rather than lipidomics. In addition to the information that can be obtained from searching in academic databases, it is also possible to consult specialized compendiums available online, such as vesiclepedia [31]. A search in this compendium can also show how information on bacterial OMVs lipidomics is very limited and suggests that the data obtained during this study may be one of few attempts to characterize *E. coli* OMVs from the lipidomics point of view (see Appendix A Appendix A).

On the other hand, regardless of the relative lack of information, it is possible to make further analysis of the data if assuming that OMVs lipid composition is similar to the *E. coli* outer membrane composition, as suggested by previous studies [32]. In that case, special attention should be paid to compounds like PGs and PEs, as these are reported to be the prevalent lipid components of the *E. coli* outer membrane [21,26], particularly for the K12 strain, which is the strain from which the JC8031 strain was derived [19]. Indeed, PGs and PEs were detected in our OMVs samples, although no significant changes in lipid composition can be inferred from the results when comparing OMVs from the designed single deletion mutants against OMVs from the JC8031 strain [33]; none of the genes that were deleted during the present study (poxB, sgbE, gmhA, and allD) were reported previously as responsible of causing a significant change in the lipid composition of the bacterium and it might be assumed that they should not cause significant changes in OMVs lipid composition either.

Another important aspect to take into account is the prevalent fatty acyl chains forming part of the OMVs. In our case, the prevalent species are vaccenic acid (18:1) and palmitic acid (16:0) (Appendix A), both of them were previously reported as prevalent in *E. coli* membranes [26]. In this last report, it was also stressed that palmitic acid was slightly more prevalent in the outer membrane. Our results show that palmitic acid is indeed slightly more prevalent when analyzed in the PEs context, but a more quantitative analysis is necessary to obtain conclusive data. This kind of analysis will be particularly important considering that OMVs are expected to be relatively rigid, thanks to a higher content of saturated fatty acyl chains [32].

The lipid profile of *E. coli* OMVs presented an unexpected major component in glycerolipids, while expected major components glycerophospholipids were found in second place. While glycerophospholipids are usually present in the outer membrane in large proportions, it is not uncommon for membranes composed of glycerolipids [34]. The lack of phosphorus in membrane lipids is associated with the depletion of phosphorus in the growth media [35]. In this case, LB culture media presented a rich nutritious environment for cell growth which provided phosphorous. This implies that the small participation of glycerophospholipids in the *E. coli* OMV lipid profile is not due to phosphorus availability. It was not possible to determine if the OMV lipid profile correlates with outer membrane profile in whole *E. coli* cells, or if vesicle formation was preferential in regions low in phosphorus or with higher amounts of glycerolipids. Additionally, proteomic studies have found an enhanced carryover of inner membrane to the generated vesicles [5]. This could have implications for the engineering of membrane proteins for interaction with host cells and could partly account for the unexpected high glycerolipids content in the isolated OMVs. Further work is needed to analyze differences in OMVs from the originating outer membrane and the effect of phosphorus availability on the lipid profile.

Advances in genomics have enabled the identification of an organism’s full gene set. Subsequent genome annotation in *E. coli* facilitates the reconstruction of enzymatic reaction networks. These networks represent the potential metabolic states a cell can achieve. However, a significant limitation in metabolic modeling stems from the completeness of genome annotation. Gaps or inaccuracies in annotation can lead to disconnected reactions within the model, thereby altering predicted metabolic flux distributions and potentially causing blocked pathways or preventing in silico growth. To address such limitations in *E. coli*, progressively refined metabolic network models have been developed [18]. These models have expanded in scope, incorporating an increasing number of metabolites and reactions, and extending beyond core metabolism to include secondary metabolism and other cellular functions. Despite these advancements, some reaction discontinuities within the network persist. Model refinement algorithms, such as GrowMatch, facilitate the integration of experimental data to improve the consistency between in silico predictions and biological reality. When applied to the model of a high vesicle-producing *E. coli* strain, GrowMatch indicated a reduction in the disparity between model predictions and experimental findings. While this refinement cannot fully rectify all inconsistencies in predicted growth capabilities, it brings the model’s performance closer to available data and observed growth patterns.

The formation of OMVs is driven by structural dynamics of the cell envelope. Certain metabolic pathways contribute to OMV production, primarily by altering phospholipid composition, which can destabilize the envelope and increase vesiculation [11]. This study centered on the metabolic networks that generate essential OMV building blocks: lipidA and glycerophospholipids, among other cell envelope precursors. Modeling the simultaneous overproduction of these multiple precursors presents a significant challenge due to the dense connectivity within metabolic networks, complicating objective function optimization. This contrasts with studies targeting a single metabolite, where the impact of network modifications is generally easier to trace. To improve the predictive power of the metabolic model, additional information is required as input to further constrain the model and contextualize it to experimental data. Future work should include the improvement of the model through the integration of lipidomic and transcriptomic data. Integration of lipidomic data is challenging as lipids are usually expressed in generic metabolites in metabolic models, and experimental data are not easily associated with the network.

## 4. Materials and Methods

### 4.1. Vesicle Recovery and Quantification

A high vesicle yield *E. coli* JC8031 strain was selected as the base strain (also referred to as WT strain). *E. coli* JC8031 is a tol/pal mutant [19]. This strain was modeled in silico and used for experimental procedures, vesicle recovery, and gene editing. *E. coli* JC8031 was transformed with a GFP production plasmid with an ampicillin resistance cassette for selection.

*E. coli* harboring a GFP plasmid was cultured in 50 mL of Luria–Bertani (LB) medium supplemented with ampicillin (100 ug/mL final concentration for plasmid selection). Cultures were incubated for 24 h at 37 °C with shaking at 250 rpm to harvest outer membrane vesicles (OMVs) during the late logarithmic growth phase, minimizing cell debris [11].

Subsequently, the culture was transferred to 50 mL conical tubes and centrifuged at 4500 rpm at 20 °C for 30 min to pellet the cells. A volume of 30 mL of the resulting supernatant was filtered through a 0.45 μm polyamide syringe filter to remove residual cells and larger debris. The clarified filtrate was then ultracentrifuged at 50,000× *g* for 3 h at 4 °C to pellet OMVs [5].

The presence of OMVs was confirmed by observing a translucent pellet exhibiting green fluorescence under blue light. The OMV pellet was resuspended in 1 mL of phosphate-buffered saline (PBS) containing spectinomycin and stored at −80 °C until further analysis. To verify the absence of viable cells, an aliquot of the resuspended OMV solution was plated onto LB agar containing ampicillin (100 µg/mL) and checked for cell growth.

The relative quantity of outer membrane vesicles (OMVs) was estimated based on UV absorption spectroscopy. Absorbance spectra were recorded using a UV spectrophotometer over a wavelength range of 200 nm to 320 nm, with measurements taken at 2 nm intervals. OMV abundance was then determined by calculating the area under the curve (AUC) for the absorbance spectrum within this defined range. Additionally, the protein concentration in each sample was semi-quantitatively estimated using the Warburg–Christian equation using absorbance values at 280 nm for protein content and 260 nm to account for the influence of nucleic acid content [24]. Vesicle production and protein concentration were normalized to the mass of sample and OD_600nm_ to account for varying growth rates between sample groups.

### 4.2. Metabolic Network Model

The *Escherichia coli* K-12 genome-scale metabolic model, iML1515, developed by Monk et al. [18], served as the initial framework for in silico assessment of outer membrane vesicle (OMV) production and identification of optimal culture conditions. Prior to use, this model was curated to confirm the accurate representation of all necessary biomass precursors and a functional biomass growth reaction. Subsequently, the iML1515 model was further adapted to reflect the genetic background of *E. coli* strain JC8031 by incorporating its known mutations.

### 4.3. Model Refinement

Experimental data on outer membrane vesicle (OMV) production levels and growth rates for 3908 *E. coli* Keio collection knock-out mutants were obtained from Kulp et al. [11]. Flux balance analysis (FBA) [36] was employed to simulate the biomass growth rates of 150 of these mutants, which were reported by Kulp et al. [11] to exhibit a high OMV production phenotype. These FBA simulation results were then compared against the corresponding experimental growth rate data to assess the predictive consistency between our in silico model and the in vitro observations. Subsequently, subsets of this validated dataset were utilized for model-guided strain design using constraint-based optimization methods [37] to identify targets for enhancing OMV production.

Concordance between in silico growth predictions and experimental observations was categorized into four types: (1) agreement where both predicted and observed growth (Growth/Growth, GG); (2) agreement where both predicted and observed no growth (No Growth/No Growth, NGNG); (3) inconsistency where growth was predicted but not observed experimentally (Growth/No Growth, GNG); and (4) inconsistency where no growth was predicted despite experimental observation of growth (No Growth/Growth, NGG). To reconcile these discrepancies with the Keio collection growth data, the GrowMatch algorithm [20] was employed. Specifically, GNG inconsistencies were addressed by GrowMatch through the targeted exclusion of reactions from the metabolic model. Conversely, NGG inconsistencies were resolved by incorporating necessary biochemical reactions identified from an external database.

### 4.4. Model Objective Function Definition

A well-defined objective function is important for accurately simulating the metabolic state associated with outer membrane vesicle production. Optimization of this function alters the in silico metabolic flux distribution, thereby guiding the model towards a biologically meaningful phenotype. Initial attempts to define an OMV-specific objective function using established algorithms based on methodologies by Burgard and Maranas [38] yielded linear combinations of model reactions. However, these algorithmically derived functions did not align with known OMV biogenesis mechanisms, as they did not prioritize key metabolic steps in precursor synthesis, were not directly linked to OMV composition, and appeared to be over-fitted to the input data. Therefore, the objective function was described according to the expected composition of the *E. coli* OM, glycerophospholipids, and lipopolysaccharides [35] as well as the expectation for an increased phospholipid production to increase OMV formation [39]. This approach is similar to the constitution of biomass objective functions which are also key to the model [40].

### 4.5. Strain Design

To identify gene deletion targets for enhanced OMV production, the calibrated metabolic model incorporating the defined vesiculation objective function was utilized. We employed the constraint-based algorithm OptKnock [22] to predict beneficial gene knock-outs. OptKnock was configured to identify deletions that maximized the OMV production objective function while simultaneously ensuring a viable biomass production rate, thereby excluding lethal deletion strategies. These simulations were performed on the refined model under conditions mimicking LB medium, with relevant exchange reactions activated to allow metabolite uptake. The OptKnock search was constrained to a maximum of five gene deletions.

### 4.6. Genome Editing

Target genes were knocked out by complete gene deletion by using CRISPR-Cas9. Guide RNAs were designed for the genes obtained through strain design using the tools from Benchling [41] and a reference *E. coli* genome (NCBI Accession: NC_000913.3). Synthetic cassettes were designed including the N20 from gRNA design (Table 2) and two segments of 500 bp corresponding to sequence flanking the target gene to provide template availability for homology directed repair of the double-stranded DNA lesion, thereby completely eliminating the targeted gene from the bacterial genome. A system of two plasmids was employed: a plasmid including Cas9 genes and a plasmid for gRNA expression [42]. All vectors were electroporated.

### 4.7. Lipidomics

Lipidomic analysis was carried out at MetCore Uniandes (Bogota, Colombia). In total, 40 uL of type I water and 160 uL of HPLC-grade methanol were added to the lyophilized samples. Samples were resuspended in a vortex for two minutes. Then, 400 uL of MTBE was added and samples were vortexed for 60 min. A total of 250 uL of type I water was added and samples were vortexed for two minutes. Samples were centrifuged at 6190 rpm and 25 °C for 10 min. Then, 20 uL of organic phase was transferred to HPLC fixed-insert vials and diluted with 80 uL of MTBE.

Quality control samples were prepared by mixing equal volumes of metabolite extract from each sample. Quality control runs were performed to stabilize the analytic platform. Subsequent quality control runs were employed every five randomized samples.

Lipidomic analysis was implemented in an Agilent Technologies 1260 Liquid Chromatography (Santa Clara, CA, USA) system coupled to a quadrupole time of flight mass analyzer and ionization by electrospray (LC-ESI-QTOF-MS). A total of 1 uL of each sample was injected in a C8 column (InfinityLab Poroshell 120 EC-C8 (150 × 3.0 mm, 2.7 µm)) at 60 °C. A gradient elution was employed composed of 5 mM ammonium formiate in Milli-Q water (Phase A) and 5 mM of ammonium formiate in isopropanol-methanol 15:85 (Phase B) with a constant flux of 0.4 mL/min. Mass spectrometry detection was performed in ESI positive mode in full scan from 100 to 1100 *m*/*z*. Mass correction was employed throughout the analysis with two reference masses: *m*/*z* 121.509 (C5H4N4) and *m*/*z* 922,0098 (C18H18O6N3P3F24).

Lipidomic profiles were obtained by using Agilent Mass Hunter Profinder 10 software employing the Recursive Feature Extraction (RFE) with extraction conditions: 0–31 min and 10,000 counts, positive ion species: (-H, +Cl, +NH4), and no additional mass or species filters. Molecular characteristics found in the solvent blank control were eliminated. Data from alignment, deconvolution, and integration were filtered by calculation of a variation coefficient (VC) of the area in QC samples. Characteristics with CV > 20% were filtered out. Data were normalized based on vesicle sample information: quantity and sample volume.

A selection of statistically significant molecular characteristics was performed through multivariate statistical analysis (MVA) and univariate (UVA). MVA was performed using SIMCA-P + 16.0 software (Umetrics) and UVA was performed by implementing the MetaboAnalyst 4.0 tool (#). Annotation of molecular characteristics was performed using the CEU MASS MEDIATOR tools (#) by batch analysis with the following parameters: tolerance = 10 ppm; databases = LipidMaps; Metabolites = OnlyLipids; input masses mode= *m*/*z* masses; ionization mode = positive; adducts = (M + H, M + Cl, M + NH4). Additionally, molecular formulas were generated for statistically significant molecular characteristics by using the Agilent MassHunter Qualitative 10 software with positive ions = (-H, +Cl, +NH4) and elements= (H, C, O, N, P) as parameters.

### 4.8. Statistical Analysis

Data were analysed using GraphPad Prism version 9.3.1 (GraphPad Software, Boston, MA, USA). A one-way ANOVA was used to estimate the statistical significance of differences within groups. Sidak’s multiple comparison test was used to evaluate statistical differences with wild-type group JC8031. Differences were considered significant at *p*-values of < 0.05.

## 5. Conclusions

In this study, we generated four single deletion mutants through in silico metabolic network model strain design and CRISPR-Cas9 genome editing (ΔpoxB, ΔsgbE, ΔgmhA, ΔallD). The four designed strains showed an enhanced vesicle production of up to 10 times that of the wild-type JC8031 strain while maintaining cell growth, while only three displayed a greater protein concentration (ΔpoxB, ΔsgbE, ΔallD). This is a step ahead to obtain OMVs in a large-scale setting and shows that it is plausible to use metabolic network modelling the complex vesicle formation process while targeting single genes.

Lipidomic analysis though LC-ESI-QTOF-MS showed that there were no significant changes in the lipid profile that separated the four mutants and WT in the different groups. The OMV lipid profile remained stable across strains. Recovered *E. coli* OMVs displayed a lack of phosphorus in their composition as glycerolipids were the majority component while glycerophospholipids were present in a lesser quantity. Knowledge of the *E. coli* OMV lipid profile and lack of alterations in different single deletion strains is useful to employ OMVs as therapeutic alternatives and drug transport and delivery.

Strain design using metabolic network models encounters difficulties when the target for production is a complex structure, such as OMVs, instead of a single metabolite. Integration of multiple source omic data should be used to improve the model.

## Figures and Tables

**Figure 1 ijms-26-06714-f001:**
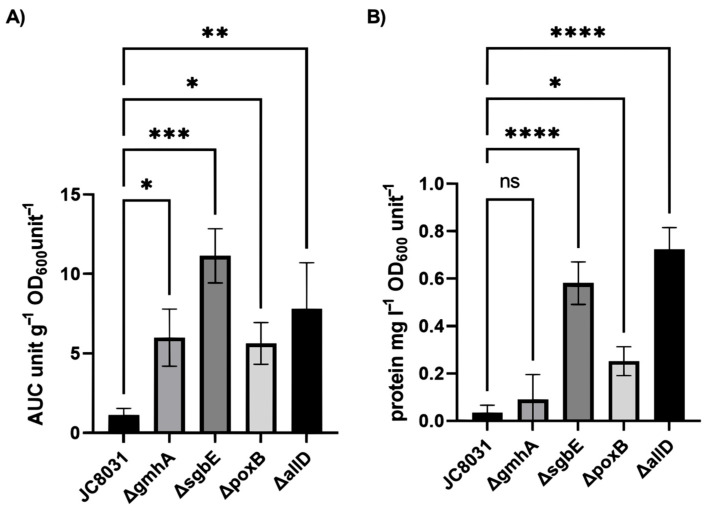
(**A**) Spectrophotometric determination of mean vesicle production for *E. coli* JC8031 and the four constructed mutants, Δ*poxB*, Δ*sgbE*, Δ*gmhA*, Δ*allD,* normalized to mass of sample and OD_600_. (**B**) Mean protein concentration according to the Warburg–Christian equation. (one-way Anova presented *p* < 0.0001; Asterisks show Sidak’s multiple comparison test results using Sidak correction, ns: *p* > 0.05; *: *p* < 0.05; **: *p* < 0.01; ***: *p* < 0.001; ****: *p* < 0.0001, all groups have n = 3). Error bars were obtained from three biological replicates.

**Figure 2 ijms-26-06714-f002:**
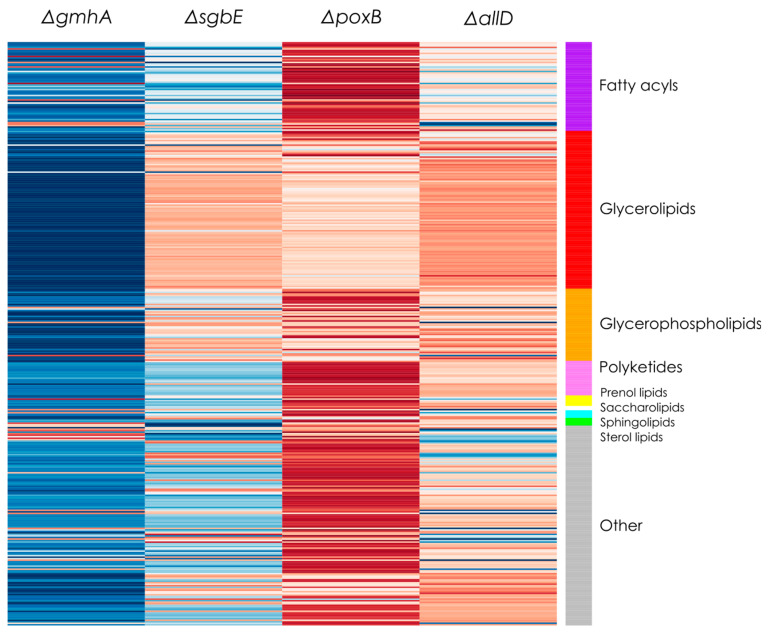
Lipid groups percentage change of four constructed *E. coli* mutants, Δ*poxB*, Δ*sgbE*, Δ*gmhA*, and Δ*allD* compared to WT *E. coli* JC8031 strain. Darker shades of blue indicate a higher lipid group content for the WT strain (negative change in relative abundance), darker shades of red indicate higher lipid group content in the respective mutant (positive change in relative abundance).

**Figure 3 ijms-26-06714-f003:**
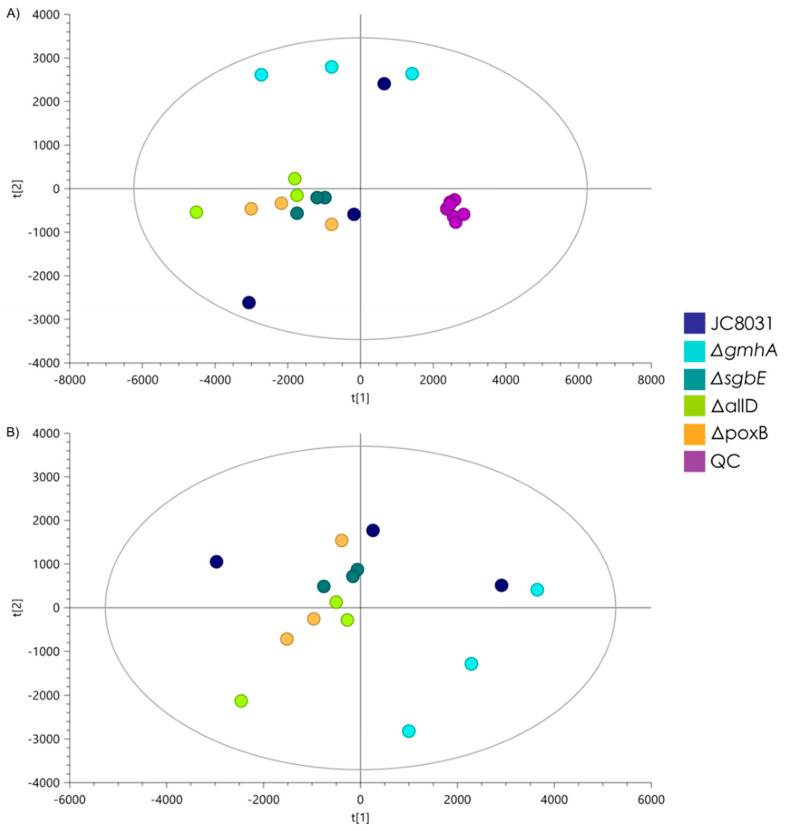
PCA and PLS-DA models for lipidomic analysis by LC-ESI-QTOF-MS of OMVs from *E. coli* JC8031 and Δ*poxB*, Δ*sgbE*, Δ*gmhA*, and Δ*allD* mutants. (**A**) PCA score plot and quality control samples (QC). R^2^ (cum): 0.954, Q^2^ (cum): 0.906; (**B**) PLS-DA model, R^2^ (cum): 0.885, Q^2^ (cum): −0.159, CV-ANOVA = 1.

**Table 1 ijms-26-06714-t001:** Gene deletions obtained through strain design using OptKnock.

Gene	Keio Code	Description
*poxB*	B0871	Peripheral membrane protein
*sgbE*	B3583	L-ribulose-5-phosphate-4-epimerase
*gmhA*	B0222	Phosphoheptose isomerase
*allD*	B0517	Ureodiglycerol dehydrogenation

**Table 2 ijms-26-06714-t002:** CRISPR-Cas9 gRNA design.

Target Gene	N20	Platform	Cassette Size	Plasmid Cut Sites
*gmhA*	5′ CGTGATCAAAGCGATCGCAG 3′	pTarget [42]	1133 bp	XbaI, AvrII
*sgbE*	5′ CATCGGCGCTCACAGCAAGG 3′	pTarget [42]	1133 bp	XbaI, AvrII
*allD*	5′ GCGACTACCGTACAGGCATG 3′	pTarget [42]	1133 bp	XbaI, AvrII
*poxB*	5′ GCGACTACCGTACAGGCATG 3′	pTarget [42]	1133 bp	XbaI, AvrII

## Data Availability

We included all data in the manuscript and Appendix A.

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
