# Peer review of "Enhanced Outer Membrane Vesicle Production in Escherichia coli: From Metabolic Network Model to Designed Strain Lipidomic Profile"

_ijms, 2025, doi:10.3390/ijms26146714_

Round 1
Reviewer 1 Report
Comments and Suggestions for Authors
I have read through the manuscript and found it to be a valuable contribution to the field. However, I believe the manuscript could benefit from some revisions.
Please check major revision suggestions below:
- Figure 1: I would recommend the authors to include WT’s and the four Knock-out strains’ growth curves (OD600 nm measurements shown over time).
- The method used to quantify OMV production is limited. I would suggest the authors to use a complementary method to analyse OMV production as, for instance: perform a SDS-PAGE analysis of OMVs isolated from each strain and measure bands at ~37 kDa (OmpF, OmpC, and OmpA) or calculate the ratio lyophilized final product amount (dry weight) referred to the corresponding cellular pellet obtained after first centrifugation (wet cell weight).
Please check minor revision suggestions below:
- Escherichia coli should be in italics around the manuscript.
- Figure 2: please, include a color bar indicating fold change to enhance its clarity.
- Line 255. Delete “Compared to detergent”.
- Line 351. 50,000G à 50,000 x g
- Lines 347, 353, 358, 361. 50mL à 50 mL; 1000ul à 1000 µL; 10ul à 10 µL; etc. (blank space)
- Line 363. OD600 à OD 600nm
- Line 421. Please, clarify which method was employed to introduce the vectors into bacteria.
- Line 469. Delete from line 469 to line 488.
- Line 543. Please, delete the numbered list from reference’s section.

Author Response
Rebuttal letter
Dear Editor:
First of all we modified all the document according to the requirements received regarding paragraphs similar to other publications
Then, we considered all comments from the reviewers and Carried out the modifications accordingly. In the following paragraphs an detailed explanation is given:
Please check major revision suggestions below:
- Figure 1: I would recommend the authors to include WT’s and the four Knock-out strains’ growth curves (OD600 nm measurements shown over time).
We agreed with the reviewer so now growth curves are included in the supplementary material and it is commented in the manuscript (See Page 4, L 201-202 and supplementary material)
- The method used to quantify OMV production is limited. I would suggest the authors to use a complementary method to analyse OMV production as, for instance: perform a SDS-PAGE analysis of OMVs isolated from each strain and measure bands at ~37 kDa (OmpF, OmpC, and OmpA) or calculate the ratio lyophilized final product amount (dry weight) referred to the corresponding cellular pellet obtained after first centrifugation (wet cell weight).
We carried out SDS PAGE for all mutants but finding no interesting differences by measuring the bands (Please see P. 4 l. 204 and 205 and supplementary material).
Please check minor revision suggestions below:
- Escherichia coli should be in italics around the manuscript.
All were corrected
- Figure 2: please, include a color bar indicating fold change to enhance its clarity.
The color bar was already added to the figure on the right side
- Line 255. Delete “Compared to detergent”.
It is already deleted
- Line 351. 50,000G à 50,000 x g
It was modified
- Lines 347, 353, 358, 361. 50mL à 50 mL; 1000ul à 1000 µL; 10ul à 10 µL; etc. (blank space)
It was corrected
- Line 363. OD600 à OD 600nm
It was corrected
- Line 421. Please, clarify which method was employed to introduce the vectors into bacteria.
It is already added
- Line 469. Delete from line 469 to line 488.
Corrected
- Line 543. Please, delete the numbered list from reference’s section.
Numbers were deleted
Rebuttal letter second Reviewer
Dear Editor:
We considered all comments from the reviewers and Carried out the modifications accordingly. In the following paragraphs an detailed explanation is given:
A well-organized and creative combination of in silico metabolic modeling and CRISPR-Cas9-based genome editing is presented in the manuscript "Enhanced Outer Membrane Vesicle Production in Escherichia coli: from Metabolic Network Model to Designed Strain Lipidomic Profile" in order to maximize OMV production. The study offers important initial lipidomic insights on OMVs produced from E. coli and is unique in its use of genome-scale modeling for OMV yield increase. One advantage of computational predictions is their experimental confirmation, which lays the groundwork for potential therapeutic uses in the future.
To improve the scientific rigor and translational relevance, a few crucial issues should be addressed. Proteomic study of OMVs, more thorough statistical interpretation of lipidomic data, more thorough discussion of gene-specific effects, and the incorporation of more reliable vesicle measurement techniques are some of these. By addressing these issues, the manuscript's credibility and effect will be greatly increased.
We thank the reviewer for this valuable suggestion. We agree that a comprehensive proteomic characterization of OMVs would further enhance the biological insight and translational potential of our findings. However, we would like to clarify that such an analysis would require a considerably broader experimental scope involving additional sample preparation protocols, instrumentation, and bioinformatic pipelines that were beyond the aims and resources of the current study. Our present work was focused on demonstrating the feasibility of enhancing OMV production through metabolic modeling and CRISPR-Cas9 strain design, alongside an initial lipidomic characterization. We envision that future studies will expand on this foundation to include proteomic analysis of OMVs from both the wild-type and engineered strains to better understand the potential implications of gene deletions on OMV cargo composition and functionality.
1:Lines 194–205: Lipidomics lacks a significant statistical depth.
Although PCA and PLS-DA were done, no significant lipids were found. Authors should think about more sensitive or tailored lipidomic techniques.
We appreciate the reviewer’s observation. In response, we performed additional statistical analyses, including both alternative univariate and multivariate approaches, to further explore potential differences in lipid composition. These complementary analyses consistently confirmed our original findings: no statistically significant lipid features were identified that differentiate the OMVs of the engineered strains from those of the wild-type. While more targeted or sensitive lipidomic techniques may provide additional insight, our results suggest that, under the current experimental conditions and analytical resolution, the overall OMV lipid composition remains stable despite gene deletions.
2: There are no biological replicates or ranges of variation discussed (results in Figs 1 and 2):
The results are reported statistically significant, however there is no mention of biological replicate consistency or standard deviations.
Fig 1 caption now explains the biological replicas
3: Limited investigation of downstream application potential (Discussion & Conclusion): While therapeutic usage is mentioned, no functional test or structural stability evidence is provided to substantiate these assertions.
We appreciate the reviewer’s observation regarding the downstream application potential of the produced OMVs. Indeed, the potential use of OMVs in therapeutic applications such as drug delivery or vaccine platforms is an exciting avenue that we briefly discuss in light of their enhanced production. However, we would like to clarify that the primary goal of this study was to demonstrate the feasibility of using metabolic network modeling for strain design and to evaluate its impact on OMV yield and lipid composition.
Functional assays or detailed structural stability analyses (e.g., zeta potential, vesicle integrity under stress conditions, biological cargo delivery) would require a significantly broader experimental framework, including tailored experimental setups and additional resources. These important aspects are currently being explored in follow-up studies aimed at evaluating the bioactivity and robustness of the OMVs generated by the engineered strains.
4: Section 2.2: Weak justification for specific gene knockouts beyond in silico prediction .
The rationale lacks biochemical/mechanistic explanations for why the deletions should increase OMV production.
Now the section is extended with a deeper explanation
5: Lines 118-127: No growth curve or biomass data are displayed .
The authors indicate sustained growth but do not provide comparable growth curves for wild-type and mutant strains.
Growth rates are now displayed on the supplementary material and mentioned on the manuscript
Minor comments:
Line 17: Change "gives OMVs a high potential" to "confers OMVs with high potential".
It is now corrected
Line 19: Clarify “Single-gene mutants of the hypervesiculating strain” — better specify “derived from E. coli JC8031”.
Now it is modified
Line 25: “...did not change significantly...” → consider quantifying what constitutes significance.
Now we added an explanation
Line 33: “By shipping away...” sounds informal. Consider "By secreting".
Now it is corrected
Line 41: Use “can be engineered to target” instead of “can be leveraged to target”.
Now it is corrected
Line 62: Mention of “metabolic pathways involved... yet to be elucidated” could use a citation for emphasis.
It was modified
Line 92: Title "Results" — add subsection numbering throughout for consistency.
Subsections numbers were added
Line 105: Clarify “extracellular space” — maybe specify medium or periplasm-external.
It was modified
Line 135: Indicate how CRISPR deletions were confirmed (PCR gel images? Sequencing?).
PCR results confirmed the complete deletion of the selected gene in each of the four mutants. (Please see L. 224)
Line 148: Add concentration units for protein — μg/mL or mg/mL.
Concentration is now explained
Line 157: Expand Sidak test footnote (e.g., “Multiple comparisons performed using Sidak correction, n=3”).
It was expanded
Figure 1: Label error bars and replicate numbers in figure caption.
Done
Line 175: Typo — “indicatee” should be “indicate”.
Modified
Line 182: “Additionally, other OMV compounds remained unidentified” — vague. Any estimate or reason?
Reason added
Line 204: Replace "assumed" with “suggested” for scientific neutrality.
It was replaced
Line 254: “material stability” could be better phrased as “structural stability”.
It was modified
Line 273: “This result is consistent with…” — specify if same species/conditions.
This result is consistent with a previous study where all possible single gene deletions, in E. coli, were analyzed with respect to their effect on the membranes lipid composition
Line 287: “Relatively rigid” — quantify or cite previous modulus values if available.
Not available modulus
Line 298: “...matches the outer membrane profile...” — unclear. Suggest “correlates with outer membrane lipid composition”.
Modified
Line 321: Sentence too long and vague. Consider breaking or clarifying the link between vesicle modeling and metabolomics data.
modified

Reviewer 2 Report
Comments and Suggestions for Authors
Journal: IJMS (ISSN 1422-0067)
Manuscript ID: ijms-3651254
Type: Article
Title: Enhanced Outer Membrane Vesicle Production in Escherichia coli: from Metabolic Network Model to Designed Strain Lipidomic Profil
A well-organized and creative combination of in silico metabolic modeling and CRISPR-Cas9-based genome editing is presented in the manuscript "Enhanced Outer Membrane Vesicle Production in Escherichia coli: from Metabolic Network Model to Designed Strain Lipidomic Profile" in order to maximize OMV production. The study offers important initial lipidomic insights on OMVs produced from E. coli and is unique in its use of genome-scale modeling for OMV yield increase. One advantage of computational predictions is their experimental confirmation, which lays the groundwork for potential therapeutic uses in the future.
To improve the scientific rigor and translational relevance, a few crucial issues should be addressed. Proteomic study of OMVs, more thorough statistical interpretation of lipidomic data, more thorough discussion of gene-specific effects, and the incorporation of more reliable vesicle measurement techniques are some of these. By addressing these issues, the manuscript's credibility and effect will be greatly increased.
Typo error in the Title “profil” should be profile
1:Lines 194–205: Lipidomics lacks a significant statistical depth.
Although PCA and PLS-DA were done, no significant lipids were found. Authors should think about more sensitive or tailored lipidomic techniques.
2: There are no biological replicates or ranges of variation discussed (results in Figs 1 and 2):
The results are reported statistically significant, however there is no mention of biological replicate consistency or standard deviations.
3: Limited investigation of downstream application potential (Discussion & Conclusion): While therapeutic usage is mentioned, no functional test or structural stability evidence is provided to substantiate these assertions.
4: Section 2.2: Weak justification for specific gene knockouts beyond in silico prediction .
The rationale lacks biochemical/mechanistic explanations for why the deletions should increase OMV production.
5: Lines 118-127: No growth curve or biomass data are displayed .
The authors indicate sustained growth but do not provide comparable growth curves for wild-type and mutant strains.
Minor comments:
Line 17: Change "gives OMVs a high potential" to "confers OMVs with high potential".
Line 19: Clarify “Single-gene mutants of the hypervesiculating strain” — better specify “derived from E. coli JC8031”.
Line 25: “...did not change significantly...” → consider quantifying what constitutes significance.
Line 33: “By shipping away...” sounds informal. Consider "By secreting".
Line 41: Use “can be engineered to target” instead of “can be leveraged to target”.
Line 62: Mention of “metabolic pathways involved... yet to be elucidated” could use a citation for emphasis.
Line 92: Title "Results" — add subsection numbering throughout for consistency.
Line 105: Clarify “extracellular space” — maybe specify medium or periplasm-external.
Line 135: Indicate how CRISPR deletions were confirmed (PCR gel images? Sequencing?).
Line 148: Add concentration units for protein — μg/mL or mg/mL.
Line 157: Expand Sidak test footnote (e.g., “Multiple comparisons performed using Sidak correction, n=3”).
Figure 1: Label error bars and replicate numbers in figure caption.
Line 175: Typo — “indicatee” should be “indicate”.
Line 182: “Additionally, other OMV compounds remained unidentified” — vague. Any estimate or reason?
Line 204: Replace "assumed" with “suggested” for scientific neutrality.
Line 254: “material stability” could be better phrased as “structural stability”.
Line 273: “This result is consistent with…” — specify if same species/conditions.
Line 287: “Relatively rigid” — quantify or cite previous modulus values if available.
Line 298: “...matches the outer membrane profile...” — unclear. Suggest “correlates with outer membrane lipid composition”.
Line 321: Sentence too long and vague. Consider breaking or clarifying the link between vesicle modeling and metabolomics data.
Author Response

(The authors gave the same response as above.)

Round 2
Reviewer 2 Report
Comments and Suggestions for Authors
The authors have adequately addressed all reviewer comments, and the manuscript is now suitable for acceptance in its present form.